# Environmental enrichment increases transcriptional and epigenetic differentiation between mouse dorsal and ventral dentate gyrus

Tie-Yuan Zhang [1,2,3], Christopher L. Keown[4], Xianglan Wen[1,2,3], Junhao Li[4], Dulcie A. Vousden[5], Christoph Anacker[1,2,3], Urvashi Bhattacharyya[4], Richard Ryan[1,2,3], Josie Diorio[1,2,3], Nicholas O'Toole[1,2,3], Jason P. Lerch[5], Eran A. Mukamel [4] & Michael J. Meaney[1,2,3,6]

Early life experience influences stress reactivity and mental health through effects on cognitive-emotional functions that are, in part, linked to gene expression in the dorsal and ventral hippocampus. The hippocampal dentate gyrus (DG) is a major site for experience-dependent plasticity associated with sustained transcriptional alterations, potentially mediated by epigenetic modifications. Here, we report comprehensive DNA methylome, hydroxymethylome and transcriptome data sets from mouse dorsal and ventral DG. We find genome-wide transcriptional and methylation differences between dorsal and ventral DG, including at key developmental transcriptional factors. Peripubertal environmental enrichment increases hippocampal volume and enhances dorsal DG-specific differences in gene expression. Enrichment also enhances dorsal-ventral differences in DNA methylation, including at binding sites of the transcription factor NeuroD1, a regulator of adult neuro-genesis. These results indicate a dorsal-ventral asymmetry in transcription and methylation that parallels well-known functional and anatomical differences, and that may be enhanced by environmental enrichment.

[1] Sackler Program for Epigenetics and Psychobiology, McGill University, Montréal H4H 1R3, Canada. [2] Ludmer Centre for Neuroinformatics and Mental Health, McGill University, Montréal H4H 1R3, Canada. [3] Department of Psychiatry, Douglas Mental Health University Institute, McGill University, 6875 boul. Lasalle, Montréal H4H 1R3, Canada. [4] Department of Cognitive Science, University of California, 9500 Gilman Dr., La Jolla, San Diego 92093 CA, USA. [5] Department of Medical Biophysics, The Hospital for Sick Children, University of Toronto, Toronto M5G 1X8, Canada. [6] Singapore Institute for Clinical Sciences, Singapore 117609, Singapore. Tie-Yuan Zhang and Christopher L. Keown contributed equally to this work. Correspondence and requests for materials should be addressed to T.-Y.Z. (email: tieyuan.zhang@mcgill.ca) or to E.A.M. (email: emukamel@ucsd.edu)

The hippocampus is implicated in learning and memory, as well as the processing of emotional stimuli and regulation of stress responses. Dorsal and ventral hippocampal regions exhibit distinct connectivity and functional roles despite similar cell type composition[1]. The dorsal hippocampus, corresponding to the posterior hippocampus in primates, associates closely with cognitive functions and age-related cognitive impairments. In contrast, the ventral hippocampus, (anterior region in primates) is implicated in the regulation of emotional states and vulnerability for affective disorders. This functional specialization is reflected in patterns of gene expression. Gene expression in the dorsal hippocampus correlates with that in cortical regions involved in information processing, while genes expressed in the ventral hippocampus correlate with expression in limbic regions involved in emotion and stress[1]. In addition, transcriptomic analysis reveals profound molecular differences, even within a uniform cell type population such as dorsal and ventral DG granule cells[2]. Epigenetic regulation may underlie these molecular differences and is also a potential mechanism for environmental influences on hippocampal development[3].

Early life experience has a profound, lifelong impact on emotional health due, in part, to environmental factors that influence gene expression in brain regions critical for cognitive-emotional stress responses. Epigenetic mechanisms such as DNA methylation, demethylation, and chromatin remodeling, have been linked to adult neurogenesis in the DG[4] and to neuronal plasticity underlying learning and memory[5,6]. DNA methylation could likewise play a role in mediating long-term effects of early life experience[7]. Epigenetic modifications of DNA and histone proteins also define tissues and cell types during development[8–10], complicating the interpretation of epigenomic data from heterogeneous samples.

To elucidate the role of region-specific epigenetic regulation in the DG, we generated transcriptomes and base-resolution, whole-genome DNA methylation and hydroxymethylation profiles for the dorsal and ventral DG. Our data and analyses reveal substantial asymmetries between the DNA methylomes of the two hippocampal poles, and suggest that enriched environment (EE) enhances dorsal-specific epigenomic signatures.

## Results

**Environmental enrichment promotes hippocampal neurogenesis.** Using high-resolution in vivo structural magnetic resonance imaging (MRI)[11,12], we found that hippocampal volume is enlarged in mice raised in an enriched environment (EE) compared with standard housing (SH) in both the dorsal (8.5% greater volume, $p = 0.001$, Student's $t$-test) and ventral poles (6.1%, $p = 0.039$; significant interaction between region and condition, $p = 0.017$) (Fig. 1a). EE also associates with >60% more newborn neurons labeled by 5′-bromo-2′-dexoyuridine (BrdU), a marker of proliferating cells[13], in the DG (dorsal, $p = 0.0097$;

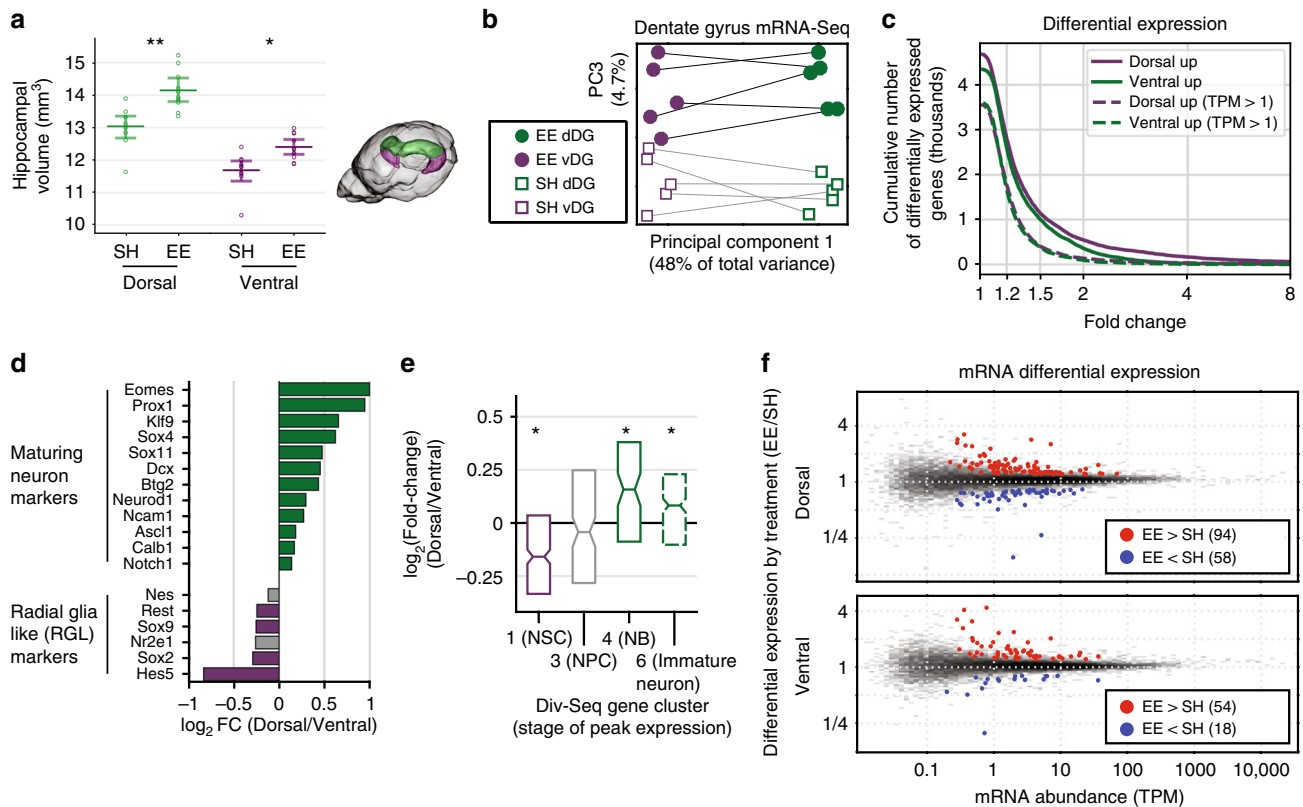

**Fig. 1** Transcriptional effects of enriched environment (EE) are greater in dorsal than ventral dentate gyrus (DG). **a** High-field structural MRI shows enlarged hippocampus in EE-treated animals. **b** DG transcriptome principal components separate dorsal and ventral samples (PC1), as well as standard housing (SH) vs. EE reared animals (PC3). Dorsal and ventral samples from the same mice are connected by lines. **c** The cumulative number of genes differentially expressed in dorsal vs. ventral DG (FDR < 0.05) as a function of the minimum expression difference cutoff. Here we consider all genes with >10 RNA-Seq read counts in ≥2 samples (solid lines), or with TPM >1 in ≥3 samples (dashed). **d** Maturing neuron and radial glia like (RGL) markers[4, 17] are enriched in dorsal and ventral DG, respectively (gray bars: not significant). **e** Clusters of genes active in RGL or immature neurons in Div-Seq data[14] are enriched in dorsal and ventral DG, respectively. NSC neuronal stem cell, NPC neural progenitor, NB neuroblast. **f** Twice as many genes are differentially expressed in EE vs. SH in dorsal compared with ventral DG

ventral, $p = 0.028$; Supplementary Fig. 1A, B). These results are consistent with previous findings that enrichment increases hippocampal volume and neurogenesis in the dentate gyrus[11,12].

**Specialization of gene expression in dorsal and ventral DG.** To address the molecular basis for the effect of EE on hippocampal function, we used RNA-Seq to profile gene expression in dorsal and ventral DG. Dentate granule cells have distinct gene expression patterns at the two poles[2], and single-nucleus transcriptome profiling has been used to link patterns of gene expression with the developmental trajectory of newborn neurons[14] and the activation of immediate early genes in a novel environment[15]. However, the impact of environmental enrichment on the specialized gene expression programs of the dorsal and ventral DG has not been examined. To increase the statistical power of our gene expression analysis and to limit variability due to single-nucleus isolation or microdissection, we performed RNA-Seq in carefully dissected whole-tissue samples of dorsal and ventral DG from five independent biological replicates in each condition (each replicate used pooled tissue from $n = 10$–12 animals; see STAR methods). Compared with microdissection-based RNA-Seq data[2], our gene expression profiles showed high correlation between samples (Spearman correlation for replicates, $r = 0.988$ compared with $r = 0.785$, Supplementary Fig. 2A–F). This level of quantitative precision in our data allowed us to comprehensively detect gene expression changes due to EE in the dorsal and ventral DG. Although our samples from whole tissue comprise multiple neuronal and glial cell types, the gene expression profiles we observed were most strongly correlated with expression from purified neurons compared to non-neuronal brain cell types, suggesting the tissue is primarily composed of neurons (Supplementary Fig. 2P)[16].

Transcriptome-wide analysis showed that dorsal-ventral differences in gene expression account for nearly half of the total data variance (Fig. 1b, Supplementary Fig. 2G). Over 28% of genes expressed in the DG were differentially expressed by region (3497 out of 12,247 genes; false discovery rate (FDR) < 0.05, TPM >1, fold-change >20%, Fig. 1c; Supplementary Data 1), including 244 genes (2%) with >2-fold difference in expression. Genes that were previously reported to show skewed expression in dorsal vs. ventral dentate granule cells[2] were similarly skewed in our data (Supplementary Fig. 2F), including dorsally enriched *Lct*, *Abcb10* and *Spata13* and ventrally enriched *Trhr*, *Grp*, and *Cpne7*. This consistency further supports the substantial contribution of granule neurons to our RNA-Seq data.

We found similar numbers of genes upregulated in the dorsal and the ventral regions. Although differential expression was widespread, the magnitude of expression differences was ~4-fold smaller than the differences between distinct cortical cell types[9] (Supplementary Fig. 2H). We found notable differences between dorsal and ventral expression of key developmental factors such as ventrally-upregulated *Nr2f1/2* and dorsally-upregulated *Epha7*. Transcription factors that mark radial glia-like (RGL) stem cells (e.g., *Sox2*, *Hes5*) were enriched in the ventral DG, whereas maturing neuron markers (e.g., *NeuroD1, DCX*) were enriched in the dorsal DG (Fig. 1d, e)[4,17], consistent with more active neurogenesis in the dorsal DG[18]. These data suggest specialized transcriptional regulation of neurogenesis in the dorsal and ventral DG.

Gene expression was more affected by EE in dorsal than ventral DG (Fig. 1b, greater separation of EE and SH samples on PC3 for dorsal than ventral, Supplementary Fig. 2I, J), and dorsal DG has twice as many differentially-expressed genes (152 dorsal, 72 ventral; FDR < 0.05 and fold change ≥ 20%; Fig. 1f; Supplementary Data 1). The 37 genes upregulated in both regions were enriched for learning and memory function and included genes induced during neuronal activation (*Junb, Arc, Fos, Npas2/4*) that play critical roles in contextual memory formation[19]. *Gadd45b* was upregulated by EE in both regions and is implicated in activity-induced demethylation of gene promoters associated with neurogenesis[19]. Overall, our transcriptome analyses based on RNA sequencing, which we validated with amplification-free digital RNA quantification (Supplementary Fig. 2K–N), are consistent with enhanced neurogenesis following EE, particularly in the dorsal DG.

**DNA methylation differences between dorsal and ventral DG.** DNA methylation is a stable epigenetic mark that could mechanistically support enduring transcriptional differences between dorsal and ventral DG and mediate the lifelong effects of early experience. Neuronal cell types show unique patterns of both CG and non-CG methylation (denoted mCG, mCH)[9], as well as hydroxymethylation (hmC)[20,21]. However, methylation differences have not been examined within relatively homogenous cell types such as dentate granule cells arrayed along the longitudinal axis of the DG. Our RNA-Seq data showed that enzymes involved in DNA methylation (Dnmt1, Dnmt3a,b) and demethylation (Tet1,2,3, Gadd45a) are enriched in the dorsal compared to the ventral pole of the DG (Supplementary Fig. 2O). To examine mCG, mCH, and hmC with single base resolution genome-wide, we performed bisulfite sequencing (MethylC-Seq) and Tet-assisted bisulfite sequencing (TAB-Seq)[22] on each of 20 samples (5 independent samples per condition from dorsal and ventral DG; 14.8-fold genome coverage per sample), a dataset unprecedented in its scale.

Each of the three forms of methylation exhibited a distinct genomic distribution in dorsal and ventral DG, leading to clear separation of dorsal and ventral samples in terms of methylation principal components (Supplementary Fig. 3A). A striking example is the locus containing *Nr2f2 (COUP-TF2)*, a developmental factor upregulated in ventral DG[2,23]. The gene body of *Nr2f2* is surrounded by a large, ~50 kbp DNA-methylation valley (DMV) that is dorsally hypomethylated in terms of mCG, mCH, and hmCG (Fig. 2a, boxes i,ii,iv). The opposite pattern, ventral hypomethylation, prevails within the gene body of the shorter isoform, *Nr2f2.2* (box iii), consistent with the strong ventral-specific expression of this gene (>4-fold). The presence of large DMVs with both dorsal and ventral hypomethylation signatures at this locus illustrates the complex, region-specific relationship between DNA methylation and gene expression. We found additional DMVs associated with differentially expressed transcription factors such as *Nr2f1*, as well as the developmental patterning factor *Pax7* (Supplementary Fig. 4).

Non-CG methylation (mCH) accumulates within neurons during post-mitotic maturation in the first 4 weeks of life in mouse frontal cortex[21] and accounts for ~25% of methylcytosines in adult mouse DG[24]. Genome-wide, we found nearly twice as much mCH in ventral compared with dorsal DG ($p < 0.01$, Fig. 2b). This finding could be explained if increased neurogenesis in dorsal DG leads to a higher proportion of immature neurons, which may lack mCH. Global mCG and hmCG levels were equivalent in dorsal and ventral DG, and EE had no effect on global methylation levels. We did not detect significant hydroxymethylation at non-CG sites, consistent with cortical neurons and embryonic stem cells[21,25].

A key advantage of whole-genome DNA methylation profiling is the ability to identify differentially methylated regions (DMRs), often far from any gene body, that mark tissue-specific gene regulatory elements[9,10]. We found ~23,000 DMRs that were hypomethylated in the dorsal relative to ventral DG[26]

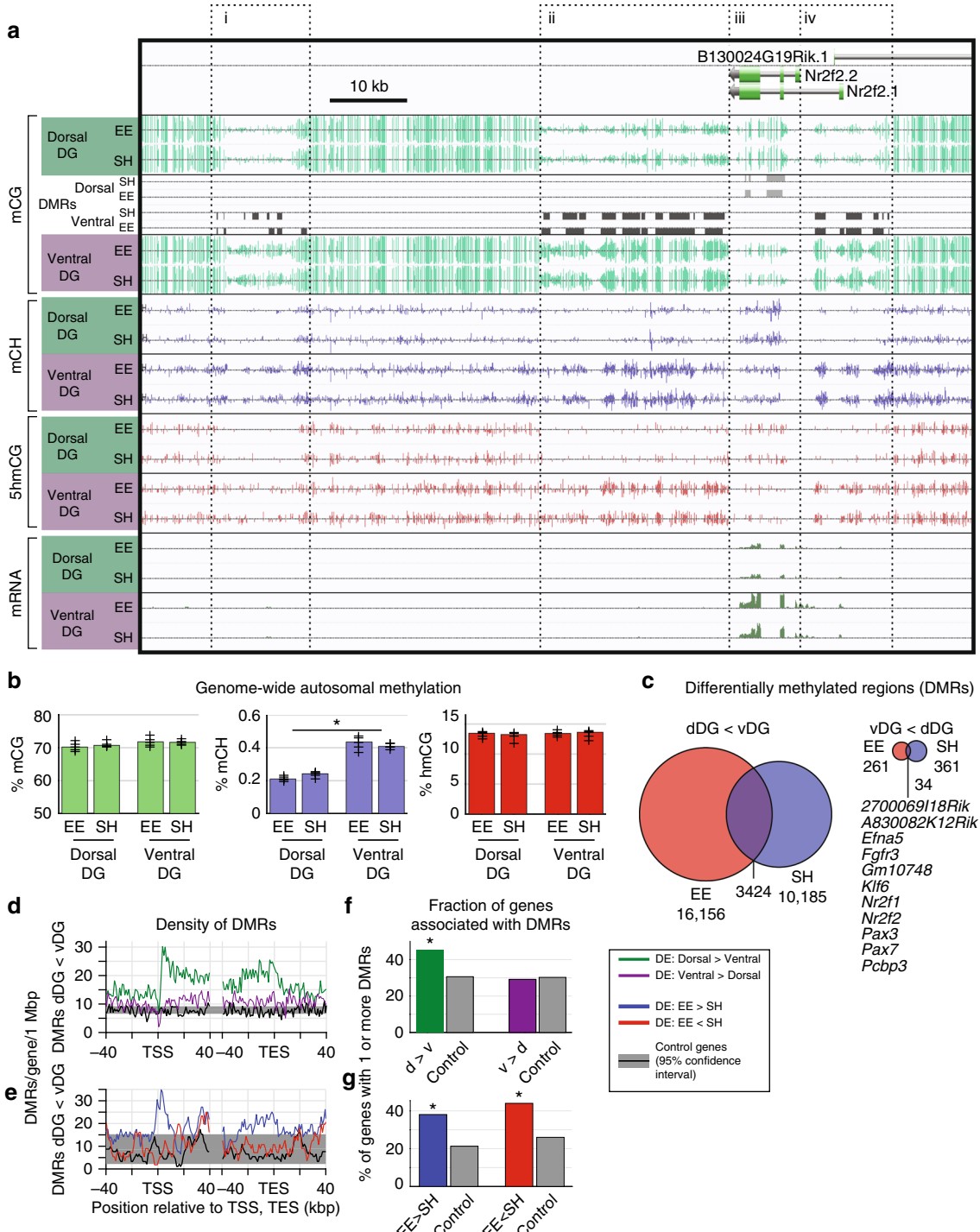

**Fig. 2** Reduced DNA methylation in dorsal dentate gyrus associated with expression differences. **a** Browser view of the locus containing development factor *Nr2f2* (*Coup-TF2*) shows bidirectional differentially methylated regions (DMRs) and corresponding differences in DNA methylation (mCG, mCH), hydroxymethylation (hmCG), and mRNA expression. **b** The genome-wide mCH level is ~50% lower in dorsal compared with ventral DG; mCG and 5hmCG did not differ ( + symbols indicate levels for individual samples). **c** The vast majority of region-specific DMRs are hypomethylated in dorsal (dDG < vDG). The smaller number of ventral hypomethylated DMRs (vDG < dDG) includes many key developmental transcription factors. **d**, **e** DMRs are enriched at differentially expressed (DE) genes. Gray shaded region: 95% confidence interval from control genes with equivalent mean expression. **f**, **g** Over half of DE genes contain no DMR

(hereafter called dorsal DMRs; mean methylation difference 26% ± 4.5% s.d.; Supplementary Fig. 3H, Supplementary Data 1), covering ~4.45 Mbp or 0.16% of the genome in total (Fig. 2c). In contrast, we found only 587 DMRs hypomethylated in ventral relative to dorsal DG (hereafter called ventral DMRs), covering 84

kbp. This strong bias, with ~40-fold more hypomethylated regions in the dorsal DG, contrasts with the balanced number of differentially expressed genes in dorsal and ventral DG (Fig. 1c, d), suggesting an asymmetric role for DNA methylation in region-specific gene regulation. Despite their small number,

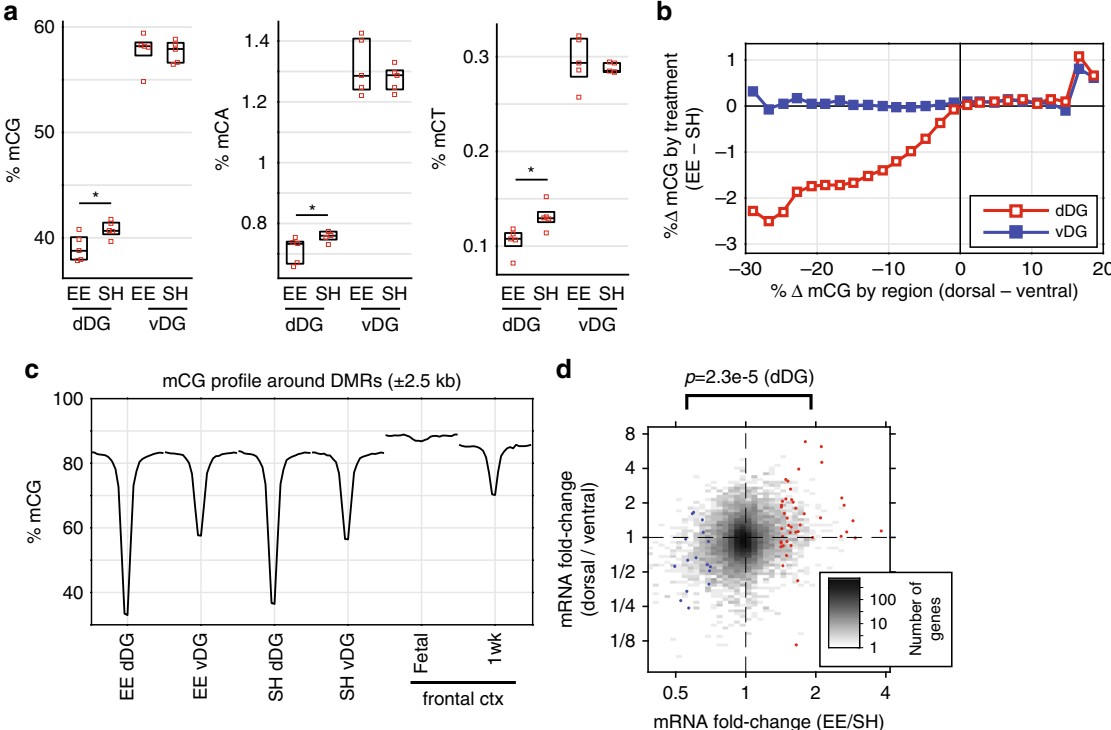

**Fig. 3** Greater dorsal-ventral differentiation of DNA methylation in enriched environment. **a** At DMRs hypomethylated in dDG, dorsal DNA methylation is lower in EE compared with SH reared animals at CG, CA and CT sites ($p < 0.05$, ANOVA). Ventral methylation is unaffected. **b** Median difference in mCG between EE and SH samples across all genomic bins (1 kbp) stratified by regional (dorsal-ventral) difference in mCG shows a strong effect in the dorsal, but not ventral, DG. **c** Mean mCG profile centered on dorsal DMRs in DG, as well as in fetal and 1 week old frontal cortex[21]. **d** Genes that are up-regulated in EE are also enriched in dDG

ventral hypomethylated DMRs marked key developmental patterning transcription factors (*Nf2f1/2*, *Pax3/7*), as well as *Efna5* and *Fgfr3* (Fig. 2c), which are linked to the proliferation, maintenance and survival of neural stem cells[27,28].

**DNA methylation correlates with repression at some genes**. CG and non-CG DNA methylation are associated with reduced gene expression, while hmC associates with increased expression, as previously observed for frontal cortical neurons[9,21] (Supplementary Fig. 3B, C, F, G). We therefore examined whether dorsal-ventral differences in methylation correlated with region-specific expression. Genes upregulated in the dorsal DG were enriched for dorsal DMRs near the transcription start site (TSS) and throughout the gene body (Fig. 2d, green curve). These DMRs were also enriched at genes that are differentially expressed in EE compared to SH treated mice (Fig. 2e). Ventrally-upregulated genes showed a significant depletion of dorsal DMRs (Fig. 2d, purple curve) and an enrichment of ventral DMRs near the TSS (Supplementary Fig. 3D). Interestingly, dorsal DMRs were also enriched at genes that were up- and down-regulated in EE, although over half of dorsal up-regulated genes, and >98.5% of ventral up-regulated genes, contained no DMRs that could explain their region-specific differential expression (Fig. 2f, g, Supplementary Fig. 3D). These DMR-independent, differentially expressed (DE) genes included some with strong (>6-fold) regional specificity (e.g., *Grp*, *Cyp26b*, Supplementary Fig. 3E). DNA methylation may thus play a targeted role in controlling regional differentiation through key transcription factors. These factors could then sustain differential expression programs in a methylation-independent manner.

**Impact of enrichment on DNA methylation in DG**. EE enhanced the epigenetic distinction between dorsal and ventral DG, leading to detection of nearly 60% more dorsal DMRs in EE (16,156 DMRs) compared with SH-reared (10,185) animals (Fig. 2c). However, only a small number of regions were statistically significant DMRs when using the same criteria to directly compare SH and EE conditions (390 hypo-methylated, 595 hyper-methylated in EE). These DMRs did not overlap between the dorsal and ventral regions. We reasoned that EE-dependent changes in DNA methylation may be enriched within the relatively abundant dorsal DMRs, and thus focused our analysis on these sites. Upon averaging over all dorsal DMRs, we found lower dorsal DNA methylation levels in EE compared with SH at both CG ($p = 0.032$) and non-CG (CA, $p = 0.049$; CT, $p = 0.017$) sites (Fig. 3a, b, Supplementary Fig. 5). Ventral DNA methylation was not significantly different between EE and SH. Dorsal DMRs were highly methylated in the fetal mouse cortex[21] and subsequently began losing methylation by one week of age (Fig. 3c). Dorsal DMRs thus mark regions that become demethylated during neuronal development. The decreased methylation of these regions in EE-reared mice is consistent with a higher proportion of immature neurons due to enhanced neurogenesis in the dorsal DG[18]. Further supporting this interpretation, we observed that most genes up-regulated by EE were also up-regulated in dorsal relative to ventral DG (Fig. 3d).

**NeuroD1 binding sites enriched at dorsal DMRs**. To address the functional significance of DG DMRs, we analyzed the enrichment of transcription factor DNA sequence motifs[29] (Fig. 4a–d). Dorsal DMRs were strongly enriched for binding motifs of NeuroD1 ($p < 10^{-200}$, hypergeometric test), a basic helix-loop-helix

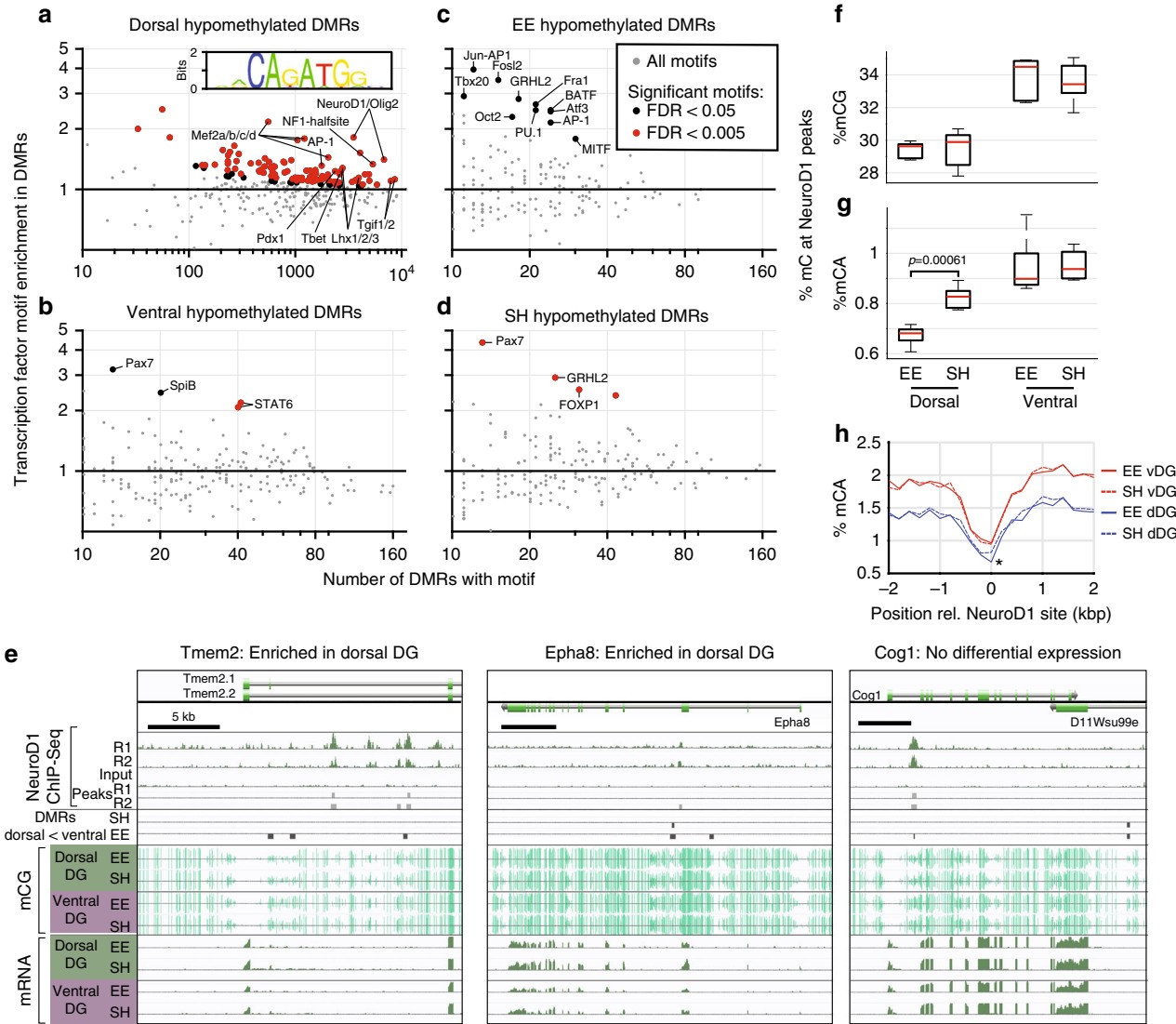

**Fig. 4** Transcription factor binding sites are enriched at DMRs. **a–d** Known transcription factor binding site sequence motifs are significantly enriched within DMRs. **a** Dorsal DMRs are enriched for motifs of developmental and neuronal differentiation TFs, including NeuroD1. Inset: sequence logo of *de novo* sequence motif matching the NeuroD1 binding motif. **b** Ventral DMRs are enriched in the TF binding site for Pax7. **c** EE DMRs are enriched for binding sites of AP-1 family immediate early genes. (**c, d**) EE and SH DMRs are enriched for GRHL2 motifs. **e** Dorsal DMRs significantly colocalize with experimentally determined binding sites of NeuroD1[32] at dorsally enriched genes (*Tmem2, Epha8*) and at some genes with no significant differential expression (*Cog1*). **f–h** mCA is significantly reduced in dDG at NeuroD1 binding sites

transcription factor that is essential for maturation of newborn hippocampal neurons[30–32]. Dorsal DMRs were also enriched in motifs of the MEF2 family of transcription factors involved in neuronal differentiation[33] (Fig. 4a). By contrast, treatment-related DMRs hypomethylated in EE relative to SH were enriched for AP-1 family motifs, indicating activation of binding sites for the immediate early genes *Fos* and *Jun* (Fig. 4c). This is consistent with our transcriptomic data (Supplementary Fig. 2I, J) showing up-regulation of *Fos* and *Fosb* in EE treated mice, and implicates AP-1 signaling as a target for the effects of EE.

Treatment-related DMRs, including both those that are hypo- and hyper-methylated in EE, are enriched with binding motifs for *Grhl2* (Fig. 4c, d), a developmental factor that may contribute to survival of neuronal progenitors via its expression in non-neuronal cells[34]. Consistent with a potential glial role, *Grhl2* mRNA is expressed at a low level in our data from dentate gyrus (0.087 ± 0.3 TPM), as well as in data from dentate granule cells[2].

To validate the motif analysis, we examined DNA methylation in the dorsal DG at experimentally determined NeuroD1 binding sites from a previous study of in vitro neuronal differentiation[32]. We found a significant overlap of NeuroD1 ChIP-Seq peaks with dorsal hypomethylated DMRs (67 peaks within 500 bp of a DMR; $p = 1.8 \times 10^{-11}$, hypergeometric test; Supplementary Data 1). A total of 48 genes contained NeuroD1 peaks collocated with a DMR (Fig. 4e). The vast majority of these genes, including *Tmem2* and *Epha8*, were significantly differentially expressed between dorsal and ventral DG (41/48); however, we also found NeuroD1 peaks overlapping DMRs in non-DE genes such as *Cog1* (Fig. 4e). Consistent with the motif enrichment analysis, we found lower mCG in dorsal compared with ventral DG at NeuroD1 ChIP-Seq peaks (Fig. 4f). Although we found no effect of EE on mCG levels at these sites, there was a significant reduction in mCA at these sites specifically in the dorsal, but not ventral, DG ($p = 0.0006$, Fig. 4g). The EE-associated differences in mCA were highly localized to the NeuroD1 binding site (Fig. 4h).

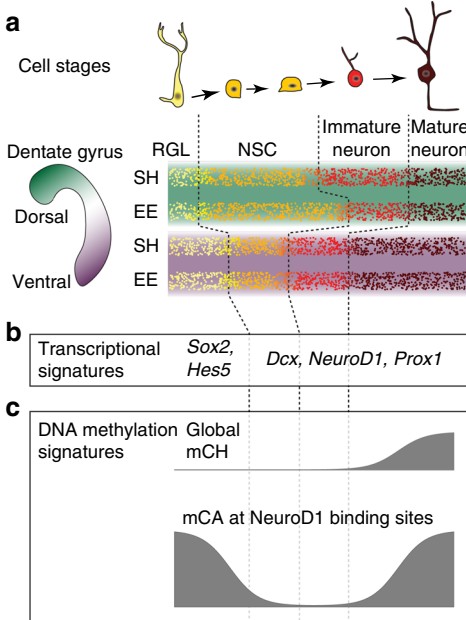

**Fig. 5** A model for epigenetic regulation of dorsal and ventral DG. **a** The cell stages occurring within the subgranular zone of the dentate gyrus are shown together with a schematic illustration of possible relative proportions consistent with our data. RGL Radial glia-like progenitor, NSC Neural stem cell. **b** Key genes associated with the RGL stage are up-regulated in ventral DG relative to dorsal DG. **c** We propose that mCH accumulates mainly in mature neurons

Thus, subregion-specific, environmental influences on dentate gyrus appear to reflect dynamic epigenetic modifications at non-CG sites within NeuroD1 transcription factor binding regions that are linked to neuroplasticity, including neurogenesis.

## Discussion

Our study integrates whole-genome, base-resolution DNA methylation and hydroxymethylation data with gene expression (RNA-Seq), in vivo structural MRI and immunohistochemistry, in a mouse model of peripubertal environmental enrichment. Environmental enrichment is a form of early experience that stably alters neural development and behavior in rodent models[35]. Using these multi-modal datasets we have identified subregion-specific transcriptomic and epigenomic influences of enriched experience in the dorsal and ventral DG. We find that the magnitude of the molecular differentiation of the dorsal and ventral hippocampus is influenced by early experience. Based on our data and analysis, we can begin to propose a unified model of epigenomic and transcriptional regulation in the DG integrating both region-specific and environmental enrichment effects (Fig. 5).

Lesion studies and connectivity profiles of the hippocampus have suggested that the dorsal hippocampus is critical for spatial cognition, whereas the ventral region is associated with emotional processing and stress responses[1]. There are substantial expression differences along the dorsal-ventral axis of the DG, as well as hippocampal subregions CA1, CA2, and CA3[2,15,36]. However, regulatory mechanisms that could support these differences remain unexamined. Our data bridge this gap, linking dorsal–ventral DNA methylation differences with transcription. For example, we identified hypomethylated regions in the ventral DG at *Pax3* and *Pax7*. These transcription factor genes restrict ventral fate in the spinal cord and could play a similar role in the hippocampus[37]. These results extend our knowledge of the

substantial epigenomic and transcriptional differences that parallel the functional specialization of the dorsal and ventral DG[1,2,14,38].

The high level of correlation ($r = 0.988$) among transcriptomes from our five independent samples allowed us to detect 3497 differentially expressed genes with high statistical confidence, far more than were previously reported in purified granule cells[2]. This illustrates that gene expression profiling in intact tissues is a valuable complement to cell type specific approaches, which may perturb the cellular transcriptome in the course of cell type purification. While the transcriptional differences we observe between dorsal and ventral DG are substantial, they are of a smaller magnitude than differences among cortical cell types (Supplementary Fig. 2H)[9]. For example, there are 4.7-fold more DE genes (using a cutoff >2-fold differential expression) when comparing cortical excitatory neurons with PV- or VIP-positive interneurons.

In contrast with the widespread differential gene expression between dorsal and ventral DG, we found a more limited number of DNA methylation and hydroxymethylation differences (mCG and hmCG). We did find a 2-fold higher abundance of mCH throughout the genome in the ventral compared with dorsal DG. Although notable DNA methylation differences at key transcription factor and ventral patterning genes were negatively correlated with gene expression, overall our data suggest that many dorsal-ventral gene expression differences cannot be directly linked to DNA methylation differences.

Adult neurogenesis in the DG is enhanced by EE[35], but the molecular mechanisms mediating this process remain unknown. Brain-derived neurotrophic factor (BDNF) is upregulated at the mRNA level in mouse hippocampus following 3–4 weeks of exposure to EE[39], while EE-induced adult neurogenesis was blocked in a heterozygous knockout ($Bdnf^{+/-}$)[40]. Similarly, mRNA for vascular endothelial growth factor (VEGF) is upregulated in hippocampus upon exposure to EE, and manipulations that increase or decrease VEGF levels cause corresponding increases and decreases in neurogenesis[41]. We did not detect differential expression of *Bdnf* or *Vegf* in the dorsal or ventral DG, suggesting these factors may be upregulated in other hippocampal regions. We did identify up-regulation in EE of mRNA for dopamine receptor D1 (*Drd1*), which is expressed in dentate granule cells[42] and gates long-term changes in synaptic strength[43,44], and the opioid neuropeptide *Penk* that is expressed in a subpopulation of DG granule cells[14]. We also found activation of immediate early genes (IEGs), consistent with increased synaptic activity. Exposure to a novel environment activates IEG transcription in DG granule cells that can be detected by single nuclei sorting followed by RNA-Seq[15]. Our data suggest IEGs are also activated by long-term exposure to an enriched environment, which includes continuous introduction of novel objects, as well as social and physical stimulation. Importantly, by performing 5-fold replicate experiments on independent biological samples, each drawn from 10 to 12 animals, we could stringently assess the reproducibility and robustness of gene expression changes.

Changes in DNA methylation can mediate long-lasting environmental effects on gene expression and behavior[3]. EE induces stable behavioral changes[45], yet the role of DNA methylation has not been examined. In our EE cohort, we observed a 31% upregulation of *Gadd45b*, involved in activity-induced DNA demethylation[19]. We found few DMRs in a direct comparison of EE and SH raised animals, indicating that individual DNA methylation changes in this paradigm may fall below the detection threshold for whole genome bisulfite sequencing. We did observe an effect of EE in modulating DNA methylation at dorsal-ventral DMRs. There were 59% more dorsal DMRs (methylation significantly lower in dorsal compared to ventral

DG) in our EE cohort compared to SH. These DMRs were enriched for binding sites of the neurodevelopmental transcription factor, NeuroD1, which is upregulated in maturing adult newborn neurons. These genomic regions also showed significantly lower methylation in EE compared to SH at CA and CT dinucleotides, suggesting an effect of early experience on a largely brain-specific form of methylation. These findings could be explained by changes in methylation within existing cells, changes in the proportion of maturing newborn neurons, or a combination of both. We also examined the role of 5-hydroxymethylcytosine (5hmC) in EE. Ten-eleven translocation (TET) family of enzymes can catalyze the conversion of 5-mC to 5-hmC[46]. Although its function is not fully understood, 5-hmC may represent an intermediate state produced during demethylation. We found 5-hmc was positively correlated with transcription, supporting the idea that 5-hmC mediates transcription.

Previous work suggests a functional distinction between the dorsal and ventral DG, and our work shows the two poles are differently affected by EE[1]. We detected 80 more differentially expressed genes in the dorsal than the ventral DG in response to EE. In addition, as noted above, the EE-reared animals showed many more dorsal DMRs (16,156) compared to SH treated animals (10,185). These regional differences may be consistent with a greater enhancement of neurogenesis by EE in the dorsal as compared to ventral DG[47].

Although our data are unprecedented in resolution and sample size, there are still some challenges to identifying the source of transcriptional and methylation changes in tissue from a heterogeneous and dynamic cell population like the DG. For example, we cannot distinguish between changes in DNA methylation occurring in a stable population of mature neurons, and changes to the proportion of immature and newborn neurons due to increased neurogenesis. Neurons in all stages of the maturation process coexist within the adult DG, and our data represent a mixture of signals from stem cells and immature and mature neurons. Similarly, the dorsal-ventral differences in DNA methylation could be driven by differences in cell type composition between the two regions, or a discrete or graded difference between the DG neurons in the dorsal and ventral poles. Here we attempted to better understand the heterogeneity of our tissue by correlating our RNA-seq data with known neuron type transcriptional profiles (Supplementary Figure 2P)[16]. Although the strongest correlation between our dDG and vDG bulk tissues was with neurons ($r = 0.89$), we also found substantial correlations with gene expression patterns in other cell types. Thus, it remains difficult to determine to what extent regional differences and EE-induced changes in cellular heterogeneity may account for our results. Future studies, including single cell assays, could address these limitations and better characterize transcription and DNA methylation in maturing newborn neurons and adult DG neurons[14,15,48,49].

Overall, our transcriptome and DNA methylation data support a model of regional and environmental effects on the molecular profile of DG neurons (Fig. 5). First, assuming only mature neurons have mCH[21] and that the mCH levels in mature dorsal and ventral dentate granule cells are similar, our finding of lower mCH in dorsal DG suggests a higher proportion of immature neurons in this region. Second, regional differences in expression of RGL and NSC markers suggest an increased proportion of NSCs in dDG and increased RGLs in vDG. This distinction is further supported by the preponderance of dorsal DMRs over ventral DMRs and their enrichment for the binding of the neuronal differentiation factor, NeuroD1. Finally, by promoting neurogenesis in the dDG, EE has the effect of further increasing the proportion of immature neurons in this region, leading to low mCG and mCA levels at dorsal DMRs and NeuroD1 binding sites.

## Methods

**Animals and environmental enrichment**. All procedures were performed in accordance with the guidelines established by the Canadian Council on Animal Care (CCAC) with protocols approved by the McGill University Facility Animal Care Committee (FACC). Male C57/Bl/6 mice were bred at the Douglas Institute to avoid transportation stress. Mice were weaned on postnatal day 22, and siblings were assigned to either standard or enriched housing conditions. Standard housed animals were raised in groups of three male mice from different mothers in a $30 \times 18$ cm cage. The enriched group contained 12 male mice, housed in a larger rectangular plexiglass cage ($78 \times 86$ cm) with a plexiglass top, which contained a variety of toys such as running wheels, a bridge, and novel objects. Toys were changed weekly. For animals in both conditions, food and water were provided ad libitum, and bedding was changed biweekly, cleaning the cages with a Peroxyguard solution. Animals remained in the respective housing conditions for eight weeks. Mice were sacrificed on age day 80 (post sexual maturation) between 1030 hours and 1200 hours. A cohort of 10 mice per housing condition was used for magnetic resonance imaging (MRI). A separate cohort was used for sequencing assays with five samples per housing condition, and each sample was composed of tissue from 10 to 12 mice. A separate cohort of male mice ($n = 20$) was used for hippocampal neurogenesis study.

**Tissue collection for MRI**. Mice were perfusion-fixed on postnatal day 80, as previously described[50]. Briefly, mice were perfused via the left ventricle using 30 ml of room-temperature (25 °C) phosphate-buffered saline (PBS) (pH 7.4), 2 mM ProHance (gadoteridol, Bracco Diagnostics Inc., Princeton, NJ), and 1 μl/ml heparin (1000 USP units/ml, Sandoz Canada Inc., Boucherville, QC) at a rate of ~1 ml/min. Next, 30 ml of 4% paraformaldehyde (PFA) in PBS containing 2 mM ProHance was infused at the same rate. After fixation, the heads, skin, ears, and lower jaw were removed and the skull was allowed to postfix in 4% PFA at 4 °C for 24 h. The samples were then placed in a solution of PBS, 2 mM ProHance, and 0.02% sodium azide (sodium trinitride, Fisher Scientific, Nepean, ON) and stored at 4 °C until imaging.

**Magnetic resonance imaging and analysis**. Anatomical whole-brain images were acquired 16 at a time using a multi-channel 7.0-T scanner and custom-built 16-coil solenoid array (Varian Inc., Palo Alto, CA)[51,52]. Brains were imaged using a T2-weighted, 3D fast spin-echo sequence at 56-micron isotropic resolution (MRI parameters: TR = 2000 ms, echo train length = 6, TEeff = 42 ms, field-of-view (FOV) = $25 \times 28 \times 14$ mm$^3$ and matrix size = $450 \times 504 \times 250$, imaging time = 11.7 h). To correct for small geometric distortions resulting from imaging in coils not in the center of the magnetic field, coil-specific MR images of precision-machined phantoms were registered to a computed tomography (CT) scan of the same phantom. The resulting distortion correcting transformations were then applied to all acquired images in a coil-specific manner.

To determine the effect of housing condition on brain anatomy, all images in the study were aligned using an automated image registration pipeline as described previously[51,53]. All registrations were performed with a combination of mni_autoreg tools[54] and Advanced Normalization Tools (ANTS)[55]. Briefly, the images were first linearly aligned using a series of global rotations, translations, scales, and shears. They were then locally aligned via an iterative nonlinear process which brings all images into precise anatomical alignment in an unbiased fashion[53,56]. The output of this automated registration process is a study-specific consensus average, representing the average anatomy of all mice in the study, along with deformation fields that encode how each individual image differs from the study average[51,53]. After registration, a manually labeled MRI atlas delineating dorsal and ventral hippocampus was aligned to the study average. This was used in combination with the deformation fields to calculate the volume of the dorsal and ventral hippocampus for each subject in the study in an automated and unbiased fashion[51,53]. The effect of housing condition on dorsal and ventral hippocampal volume was assessed using Student's $t$-tests. The interaction effect between housing condition and region on volume was assessed using a linear mixed effects model with random intercepts for each mouse using the lmerTest package[57]. Image analysis was performed using the R statistical language (R Core Team, 2016, https://www.R-project.org) and the RMINC library (https://github.com/Mouse-Imaging-Centre/RMINC). Error bars represent 95% confidence intervals.

**Immunohistochemistry**. Animals were intraperitoneally injected with BrdU (100 mg/kg, 20 mg/ml, Cat# B5002, Sigma-aldrich) twice on 2 consecutive days at postnatal day 80. After 30 days following the last injection, the animals were killed via transcardial paraformaldehyde (4% in 1 × phosphate-buffered saline) perfusion. The sliced brain sections were processed for immunohistochemistry using Anti-BrdU antibody (Abcam, Cat# ab6326, 1:400) and visualized with DAB (Cat# SK-4100, Vector Laboratories). BrdU immunoreactive cells were counted in the subgranular zone and granule cell layer region in dorsal (8–12 section, 80 μm apart, bregma −1.34 to bregma −2.30) and ventral (8–10 sections, Bregma −2.92 to Bregma −3.64)[58] hippocampus per animal under VS120 virtual slide microscope (Olympus). The number of labeled cells per dentate gyrus was statistically tested using a two-way analysis of variance (ANOVA) with housing condition and marginal region as main effects.

**Tissue collection for sequencing assays**. Tissue collection consisted of rapid removal of the brain, followed by flash freezing and storage at −80 °C. Frozen brains were sliced coronally at 200 µm thickness until reaching bregma −2.30. The brains were then removed from mounting position, rotated, and remounted to the mounting position for horizontal slicing ventral dentate tissue. Horizontal sections were sliced from interaural 3.24 to 0.92 mm[58]. A 300 µm diameter puncher was used to punch dorsal and ventral dentate gyrus region separately.

**RNA and DNA extraction**. RNA and DNA extraction were performed from the same sample using Qiagen Allprep DNA/RNA Mini kit (Qiagen, Cat# 80204.). We performed on-column DNase I treatment during RNA extraction and on-column RNaseA treatment during DNA extraction. RNA was examined by Bioanalyzer 2100 (Agilent technologies, Santa clara, USA).

**RNA-Seq collection**. The RNA libraries were prepared in McGill University and Genome Quebec Innovation Centre using Illumina TruSeq Stranded total RNA LT set (Cat# RS-122-2301, Illumina Canada Ulc.). Paired-end, 100 bp read-length RNA-seq was collected using HiSeq 2000 at a depth of 30 M sequencing.

**Validation of RNA-Seq results by digital Nanostring**. Housing differences in RNA-seq were validated with Nanostring on 48 randomly selected differentially expressed genes. In total 100 ng of tissue were sent to Jewish General Hospital (Montreal, Quebec, Canada) for expression quantification using NanoString nCounter XT-GX (NanoString Technologies, Inc., Seattle, WA, USA). Probes were designed to hit the maximum number of validated transcript variants, while minimizing the cross-reactivity of the probes. Scanned data was normalized using Nanostring-provided housekeeping genes and analyzed using nSolver Analysis Software 2.6 (NanoString Technologies, Inc., Seattle, WA, USA). Comparison of mRNA fold change between RNA-seq and NanoString shows consistent results (Supplementary Fig. 2K–N).

**TAB-Seq and MethylC-Seq**. DNA from the same samples was separated for TAB-seq and MethylC-seq library preparation. TAB-seq measures levels of 5 hydroxymethylation (5-hmC). Protection and oxidation portions of library preparation were performed in-house using the Wisegene kit as described in Yu et al.[22]. Three spike-in control DNAs, lambda DNA (Cat# D1501, Promega), 5 mC (Cat# S001, Wisegene) and 5hmC (Cat# S002, Wisegene) were added to each sample (2.5 µg of total DNA) before DNA shearing, in order to evaluate the bisulfite conversion efficiency, the protection rate of 5 hmC, and the oxidation rate of TET. In 5hmC control spike-in DNA, due to the impurity of commercial 5hmdCTP and slow oxidation of 5hmC upon exposure to air, the actual abundance of 5hmC at each cytosine site is not 100% hydroxymethylated. Therefore, we ran the same batch of 5hmC spike-in control in another bisulfite sequencing to examine its real 5hmC abundance.

Bisulfite conversion was then performed at the Genome Quebec Innovation Centre on the processed TAB-seq sample, as well as 1 µg of DNA for the MethylC-seq library. Methylated and unmethylated DNA sets (Cat# D5017, pUC19 DNA set, Zymo research) were added as spike-in controls (2 ng spike-in control in 1 µg DNA) to evaluate bisulfite conversion efficiency. The whole genome bisulfite sequencing (WGBS) libraries were prepared using NimbleGen SeqCap Epi Enrichment System (Cat# 07145519001, Roche NimbleGen, Inc.). Library amplification was done using KAPA HiFi Hotstart Uracil+DNA polymerase (Cat# KK2802, Kapa Biosystems).

**Data analysis**. All analyses were conducted in either Matlab or Python with packages including Numpy, Scipy, Pandas, Matplotlib, and Sklearn. All data were aligned to the mm10 (GRCm38) reference genome, and genes were defined using Gencode annotation vM7 level 3 transcriptome (http://www.gencodegenes.org/). Browser representations were created using AnnoJ (http://www.annoj.org)[59]. Pearson correlations were used except where stated otherwise. P-values were <0.01 unless otherwise stated.

**Differential expression**. RNA-Seq data were aligned using STAR Aligner in quantMode to obtain gene counts[60]. Differentially expressed genes were identified using generalized linear models and contrasts in EdgeR[61]. We only retained genes with counts >10 in at least two samples for the analysis. In addition, we excluded one SH sample due to high expression of the long noncoding RNA, *Xist*, which is only expressed in females. We then tested the below null hypotheses to identify differentially expressed genes by region (1) and treatment in the dorsal (2) and ventral (3) dentate gyrus. Benjamini Hochberg was used to control the false discovery rate (q < 0.05).

$$Dorsal\ SH - Ventral\ SH = Dorsal\ EE - Ventral\ EE \quad (1)$$

$$Dorsal\ EE - Dorsal\ SH = 0 \quad (2)$$

$$Ventral\ EE - Ventral\ SH = 0 \quad (3)$$

**Differential methylation analysis**. Whole genome bisulfite sequencing data were mapped using Methlypy[21]. The non-conversion rate (NCR) was estimated using a fully unmethylated phage lambda DNA spike-in. NCR was found to be low across all samples (0.43 ± 0.021%). Methylation values were corrected for the NCR using the following maximum likelihood formula, where *m* is the number of methylated base calls and *c* is the total number of base calls:

$$mC = g\left[\frac{m/c - NCR}{1 - NCR}\right],$$

$$g[x] = max[x, 0].$$

Differentially methylated regions (≥ 15% methylation difference, p < .05) at CG dinucleotides were identified using DSS[26,62]. To examine the link between differential expression and DMRs, we computed an enrichment score (the density of DMRs per gene per 1 MB) as a function of distance from the transcription start site (TSS). Enrichment scores were compared between differentially expressed and non-differentially expressed genes using a hypergeometric test.

Tet-assisted bisulfite sequencing (TAB-Seq) is a methodology for measuring genome-wide 5-hydroxymethylation that consists of three main steps: protection, the binding of a glycosyl group to hydroxymethylated cytosines; Tet oxidation, the demethylation of non-glycosylated methylated cytosines; and bisulfite treatment, conversion of all unmethylated cytosines to uracils[22]. Upon sequencing, only 5-hydroxymethylated cytosines should still be cytosines. To measure the inefficiency of each of these steps, a fully hydroxymethylated (pUC19) and a fully methylated (lambda phage) spike in are included. Corrected hydroxymethylation levels were computed using the below formula with variables $r_{TAB}$ (bisulfite non-conversion in the TAB-Seq data, estimated via Lambda DNA in the CH context), $s_{TAB}$ (non-oxidation in the TAB-Seq data, estimated using Lambda DNA in the CG context), $t_{TAB}$ (non-protection in the TAB-Seq data, estimated using pUC19), and $p_{mC}$ (the fraction of mC + hmC):

$$p_{hmC} = g\left[\frac{q_{TAB} - s_{TAB}p_{mC} - r_{TAB}(1 - p_{mC})}{1 - t_{TAB}}\right] = g\left[\frac{q_{TAB} - r_{TAB} - (s_{TAB} - r_{TAB})p_{mC}}{1 - t_{TAB}}\right].$$

Finally, we examined DMRs for enrichment of transcription factor binding sequence motifs using Homer[29]. For this analysis, sequences within 200 bp of each DMR center were included. We examined the overlap of DMRs with ChIP-Seq data[32].

**Data availability**. Raw and processed data reported in this study are available via the Gene Expression Omnibus with accession GSE95740, https://www.ncbi.nlm.nih.gov/geo/. A browser visualization of genomic data is at http://brainome.ucsd.edu/mouse_dentategyrus.

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

## Acknowledgements

We thank M.M. Behrens, J.R. Ecker, and C. Luo for helpful comments on this work. This work was supported by funding from the Hope for Depression Research Foundation (M. J.M.). E.A.M. (NIH/NINDS R00NS080911) and C.L.K. (NIMH T32 MH020002-16A) are supported by grants from the National Institutes of Health.

## Author contributions

T.Y.Z. designed the experiment, collected tissue, and developed the technique to do WGBS, TAB-seq, and RNA-seq from the same sample. T.Y.Z., C.L.K., J.L., U.B., and E.A.M. performed sequencing data analysis, with contribution from NO for RNA-Seq analysis. X.W. performed RNA/DNA extraction, prepared TAB-seq libraries and performed immunohistochemistry. D.A.V. collected MRI data and performed associated analysis. C.A., J.D. raised and sacrificed the mice. R.R. raised animals and performed BrdU experiment. J.D. supervised part of project. J.P.L. supervised the MRI project. T.Y.Z., C.L.K., E.A.M., and M.J.M. wrote the manuscript. E.A.M. and M.J.M. supervised the project. All authors read and approved the final manuscript.

## Additional information

**Competing interests:** The authors declare no competing interests.

