## [Peer Review File · Nature Communications]

Reviewers' comments:

Reviewer #1 (Remarks to the Author):

This study integrates epigenomic and transcriptional regulation along the dorsal-ventral axis of the DG and examines the influence by early experience. There are several major claims, including: (1) dorsal and ventral DG specific differences in gene expression, (2) more dorsal expression of certain proneurogenic transcription factors after EE, (3) differential methylation regions (DMRs) in dorsal vs. ventral, however this does not fully explain D vs V gene expression changes or EE changes, (4) transcription factor binding motif differences in D vs. V, including NeuroD and MEF2 binding sites, and (5) more NeuroD ChIP-seq peaks in dorsal hypomethylated DMRs which correlate with higher gene expression in dorsal DG.

Overall, the study is novel, contains robust datasets, and will be an important resource to others in the community and the wider field. Also, the paper is interesting because it gives examples of gene expression changes that correlate with DMR but it also gives examples of gene expression changes that do not correlate with DNA methylation changes. Thus, it adds to the emerging view that there is not always a 1:1 correlation with transcriptome and epigenomic changes and also highlights the importance of non-CpG methylation. However, there are several aspects that should be clarified to strengthen the conclusions.

Major Comments:

1. How stable are the gene expression and epigenomic changes in dorsal vs. ventral DG after EE? The authors should comment whether the observed gene expression and epigenomic changes are reversible, especially after return to non-enriched environment, as neurogenesis is a dynamic process.
2. What drives the hypomethylation in dorsal compared to ventral DMRs, especially at NeuroD binding sites? Is there differential TET expression or activity in D vs. V after EE?
3. The authors used only male C57Bl6 mice in the study. They should comment on the possibility of gender differences in the limitations and future work section.
4. Is Fig. 4A and 4B correctly labeled? If the NeuroD binding motif is enriched in dorsal DMR, then why is dorsal < ventral in Fig. 4A and vice versa in Fig. 4B?

Reviewer #2 (Remarks to the Author):

In this interesting manuscript, the authors analyse DNA methylation in the dorsal and ventral dentate gyrus of the hippocampus and correlate differential methylation at TSS with gene expression. Moreover, they analyse the methylation levels in response to exposure to EE conditions. Overall the experiments are well done and well controlled and the quality of the data both in terms of similarity between individual experiments and bioinformatics

analyses is quite impressive. I do agree with the authors that in most cases choosing to work with carefully dissected tissues is a better choice than using single cells, and their transcriptome data are very striking.

My major issue regards the essentially correlative nature of the findings and the descriptive nature of the study. Although I understand that this is a common problem for any study based on genome-wide analyses of epigenetics marks, it seems that the authors should have at least tried to examine the physiological implications (if any) of the methylation state and DMRs on neurogenesis and/or neuronal differentiation in both standard condition and in response to stimulation. For example, the authors could have knocked out DNMT enzymes in progenitor cells and study the effect on gene expression and neuronal differentiation at single cell levels using CRISPR technology.

Despite this important caveat, the study will certainly provide information that will be useful to many research group and will be better suited to be published as a resource, rather than as a research paper.

Reviewer #3 (Remarks to the Author):

In this manuscript the authors describe an original dataset in which whole genome bisulfite sequencing and RNA-seq was performed on dorsal and ventral poles of the hippocampus in mice. They find that pre-pubertal environmental enrichment (EE) exacerbates gene expression differences between these two regions as well as differences in DNA methylation patterns. They go on to speculate about the implications of this find and the last sentence of the abstract captures the conclusions that can be drawn completely.

The data generated are comprehensive and even though the authors are not the first to look at expression differences in hippocampus after EE, their methylation data is unique and extensive and therefore represents a valuable resource.

There are however many questions as to why differences between these two structures increase in the presence of EE and the authors seem to circle around the issue using vagueries and obscurism. My worry is that the authors merely describe changes in heterogeneity between the dorsal and ventral poles which would explain some of the weird skewing they also observe in their data. Also I wonder about the usefulness of the thresholds chosen in the various analyses and suggest a better integration of existing data. Overall, while this is a valuable resource I'm not sure how relevant their findings are and whether the data set that they have generated is even suitable to answer the questions they ask.

Specifically:

In the first line of page four the authors contrast their RNA data to existing data generated by another group and report that their data correlates replicates better. I find this an unusual way to contrast two studies especially after finding out that this previous study is a completely different type of analysis. This is a FACS based study in which purified cell populations are isolated and analyzed. The extra steps may have affected RNA quality but

circumvent effects due to sample heterogeneity. Naturally this immediately poised me to think about heterogeneity in this study. For instance the hippocampus contains two to three fold more glia than neurons thus am I correct that the bulk of the signal from any of the authors experiments measured here is differences in glial cells? In there anything known on different cellular composition in the dorsal and ventral poles that may explain why a whopping 50% of all genes are differentially expressed? Is a 20% DE cutoff with 0.05 FDR a useful cutoff in light of potential heterogeneity issues? The authors state that differences are consistent with increased neurogenesis in the dorsal pole. Is this increased neurogenesis expected to also increase differences in heterogeneity?

This is also relevant for the findings described in the DNA methylation analysis. For instance the authors describe a huge skewing of hypomethylation in DMRs in dorsal versus ventral (23000 regions vs 500). They also find less non cg methylation in dorsal vs ventral. They explanation of an asymmetric role for DNA methylation in region specific gene regulation is a typical example of the obscurism seen throughout the manuscript which cannot be more vague. Is it not more likely that differences in heterogeneity explain why in one region most of the methylation signal is higher? The authors actually describe this effect at the top of page 9 (lower overall methylation dorsally). If a region becomes more heterogeneous (for instance due to neurogenesis) all signals from specific cell types become diluted. On that note and similar to the RNA analysis, do the authors believe that the thresholds used in their analysis are workable in light of differences in heterogeneity (15% > difference)?

In the discussion after an elaborate speculation on the molecular differences between dorsal and ventral differences and the impact of EE on epigenetic signatures the authors seem to finally acknowledge that heterogeneity may be a limiting factor in their work but then revert back to speculating and continue to claim that they assess the environmental effects on DG neurons.

With the low thresholds the authors use for looking at differences in methylation I'm honestly not sure what they measure in their analysis. The authors could focus on more stringent changes (new dorsal methylation regions that could say something about the new cells generated dorsally). The authors also do not contrast their data to the existing datasets other than comparing the correlation scores of replicates to argue their data is better. Is there anything to the common changes in dorsal and ventral between SH and EE and the data previously published? Do DMRs explain differences observed in specific cell types in the previous studies? Are there key known factors to look at? Why is this not extensively cross compared?

All in all I find the data extensive and valuable as a resource albeit for a specialized audience. However I believe the analysis is trying to obscure that nothing is really yet found. I believe a more honest assessment of the data would enable the community to use the resource for what it is instead of what the authors are trying to make it and perhaps a different focus to the analysis may very well uncover a story that is fit for publication in a journal with a wide readership.

We are grateful to the editor and three reviewers for their insightful comments. Below we respond to each point raised by the reviewers. Reviewers comments are in black and our responses are in blue type.

Reviewers' comments:

Reviewer #1 (Remarks to the Author):

This study integrates epigenomic and transcriptional regulation along the dorsal-ventral axis of the DG and examines the influence by early experience. There are several major claims, including: (1) dorsal and ventral DG specific differences in gene expression, (2) more dorsal expression of certain proneurogenic transcription factors after EE, (3) differential methylation regions (DMRs) in dorsal vs. ventral, however this does not fully explain D vs V gene expression changes or EE changes, (4) transcription factor binding motif differences in D vs. V, including NeuroD and MEF2 binding sites, and (5) more NeuroD ChIP-seq peaks in dorsal hypomethylated DMRs which correlate with higher gene expression in dorsal DG.

Overall, the study is novel, contains robust datasets, and will be an important resource to others in the community and the wider field. Also, the paper is interesting because it gives examples of gene expression changes that correlate with DMR but it also gives examples of gene expression changes that do not correlate with DNA methylation changes. Thus, it adds to the emerging view that there is not always a 1:1 correlation with transcriptome and epigenomic changes and also highlights the importance of non-CpG methylation. However, there are several aspects that should be clarified to strengthen the conclusions.

We thank the reviewer for their positive comments, and we respond to the specific suggestions and criticisms below.

Major Comments:

1. How stable are the gene expression and epigenomic changes in dorsal vs. ventral DG after EE? The authors should comment whether the observed gene expression and epigenomic changes are reversible, especially after return to non-enriched environment, as neurogenesis is a dynamic process.

We agree with the reviewer that the reversibility of our reported molecular changes upon removal from EE is an important question¹. However, we believe that it is beyond the scope of the current study, although we do plan to examine it in future work. In our study, we focused on identifying region-specific molecular changes induced by EE, and particularly the role for genome wide changes in DNA methylation, a question that has not been previously examined. We consider the issue of reversibility as a matter for continued studies, which merits a different experimental design.

2. What drives the hypomethylation in dorsal compared to ventral DMRs, especially at NeuroD binding sites? Is there differential TET expression or activity in D vs. V after EE?

We agree that the mechanisms underpinning our reported methylation changes are of great interest. To address the reviewer’s question, we performed additional analyses and now include a new figure showing gene expression measured by RNA-Seq in the dorsal and ventral DG for genes directly involved with DNA methylation and/or demethylation, as well as key readers of DNA methylation. We find some evidence for enrichment of most methylating and demethylating enzymes in the dorsal compared with the ventral DG. This finding is consistent with the greater effect of environmental enrichment on DNA methylation in the dorsal versus the ventral DG. However, we cannot conclude that these expression differences in the adult DG are directly responsible for any differences in methylation, which may be impacted by dynamic changes in expression programs throughout development. Testing the mechanisms responsible for the increased number of hypomethylated DMRs in dorsal relative to ventral DG will be an interesting direction for future research. Furthermore, we added this sentence to the Results:

“Our RNA-Seq data showed that many enzymes involved in DNA methylation (Dnmt1, Dnmt3a,b) and demethylation (Tet1,2,3, Gadd45a) are enriched in the dorsal compared to the ventral pole of the DG (Supplementary Figure 20).”

Figure S20. Expression of genes associated with (de)methylation and methylation readers in dorsal and ventral DG. All genes are significantly upregulated in dorsal DG over ventral (FDR < 0.05) except *Gadd45b*, which is significantly upregulated in ventral DG, and *Gadd45g*, which is not differentially expressed. *Not significant (ns)*.

3. The authors used only male C57Bl6 mice in the study. They should comment on the possibility of gender differences in the limitations and future work section.

We agree that sex differences are critically important, as they interact with early life experience for example in stress response^{2,3}. Enriched environment has similar effects in females as males, including increased neurogenesis⁴, and sex differences in response to EE have not been demonstrated to our knowledge. For that reason, and to maximize the power of our analysis to comprehensively detect gene expression changes, we focused on male subjects in this work. We agree that examining similarities and differences in the transcriptional and epigenetic consequences of early life enrichment is an important topic for future research.

4. Is Fig. 4A and 4B correctly labeled? If the NeuroD binding motif is enriched in dorsal DMR, then why is dorsal < ventral in Fig. 4A and vice versa in Fig. 4B?

We thank the reviewer for their careful attention to detail. We reviewed the figure and confirmed that the labeling is correct. "Dorsal < ventral" indicates that CG methylation is lower at these sites in the dorsal DG compared to ventral DG, thus potentially allowing for the binding of the transcription factors indicated in the plot. To avoid confusion, we relabeled these figure panels as "Dorsal hypomethylated DMRs," etc.

Reviewer #2 (Remarks to the Author):

In this interesting manuscript, the authors analyse DNA methylation in the dorsal and ventral dentate gyrus of the hippocampus and correlate differential methylation at TSS with gene expression. Moreover, they analyse the methylation levels in response to exposure to EE conditions. Overall the experiments are well done and well controlled and the quality of the data both in terms of similarity between individual experiments and bioinformatics analyses is quite impressive. I do agree with the authors that in most cases choosing to work with carefully dissected tissues is a better choice than using single cells, and their transcriptome data are very striking.

We thank the reviewer for their appreciation of the merits of our study.

My major issue regards the essentially correlative nature of the findings and the descriptive nature of the study. Although I understand that this is a common problem for any study based on genome-wide analyses of epigenetics marks, it seems that the authors should have at least tried to examine the physiological implications (if any) of the methylation state and DMRs on neurogenesis and/or neuronal differentiation in both standard condition and in response to stimulation. For example, the authors could have knocked out DNMT enzymes in progenitor cells and study the effect on gene expression and neuronal differentiation at single cell levels using CRISPR technology. Despite this important caveat, the study will certainly provide information that will be useful to many research group and will be better suited to be published as a resource, rather than as a research paper.

We agree that experimental validation of the causal role of epigenetic modifications in regulating gene expression and behavior is essential. However, as the reviewer notes, it is nearly impossible to directly test the roles of the specific marks we observed. Such a test would require cell-type specificity, as global DNMT knockout experiments cause dramatic and lethal deficits. There is no adequate in vitro model that can capture the specific regulation of dorsal and ventral dentate gyrus neurons. Such functional validation is beyond the scope of the current study.

Reviewer #3 (Remarks to the Author):

In this manuscript the authors describe an original dataset in which whole genome bisulfite sequencing and RNA-seq was performed on dorsal and ventral poles of the hippocampus in mice. They find that pre-pubertal environmental enrichment (EE) exacerbates gene expression differences between these two regions as well as differences in DNA methylation patterns. They go on to speculate about the implications of this find and the last sentence of the abstract captures the conclusions that can be drawn completely.

The data generated are comprehensive and even though the authors are not the first to look at expression differences in hippocampus after EE, their methylation data is unique and extensive and therefore represents a valuable resource.

We appreciate the reviewer's positive comments.

There are however many questions as to why differences between these two structures increase in the presence of EE and the authors seem to circle around the issue using vagueries and obscurism. My worry is that the authors merely describe changes in heterogeneity between the dorsal and ventral poles which would explain some of the weird skewing they also observe in their data. Also I wonder about the usefulness of the thresholds chosen in the various analyses and suggest a better integration of existing data. Overall, while this is a valuable resource I'm not sure how relevant their findings are and whether the data set that they have generated is even suitable to answer the questions they ask.

Specifically:

In the first line of page four the authors contrast their RNA data to existing data generated by another group and report that their data correlates replicates better. I find this an unusual way to contrast two studies especially after finding out that this previous study is a completely different type of analysis. This is a FACS based study in which purified cell populations are isolated and analyzed. The extra steps may have affected RNA quality but circumvent effects due to sample heterogeneity.

Our study is one of the first comprehensive investigations of gene expression differences between the dorsal and ventral poles of the dentate gyrus. We therefore felt it was important to compare our findings with any previous data from these regions. Although the reviewer correctly notes that the Cembrowski et al. "Hippo-Seq" data set uses a different methodology (microdissection and FACS of single excitatory neurons), we nevertheless chose to compare with this study's results as a recent state-of-the-art characterization of dorsal-ventral differences within the dentate gyrus. We agree that the two studies and datasets have complementary strengths and weaknesses, with our data exhibiting extremely high correlation between biological replicates while the Hippo-Seq data has the advantage of coming from a purified excitatory neuron population. Nevertheless, the high variability of the Hippo-Seq data across replicates prevented the authors of that study from comprehensively surveying differential gene

expression; their paper did not include a list of statistically significant differentially expressed genes. By contrast, the high reproducibility of our data across 5 independent biological replicates allowed us to detect differential expression in thousands of genes. To clarify our interpretation we added these sentences to the Discussion:

The high level of correlation ($r = 0.988$) among transcriptomes from our five independent samples allowed us to detect 3,497 differentially expressed genes with high statistical confidence, far more than were previously reported in purified granule cells[2]. This illustrates that gene expression profiling in intact tissues is a valuable complement to cell type specific approaches, which may perturb the cellular transcriptome in the course of cell type purification.

We have also added additional comparison of our data with Hippo-Seq, as described below.

Naturally this immediately poised me to think about heterogeneity in this study. For instance the hippocampus contains two to three fold more glia than neurons thus am I correct that the bulk of the signal from any of the authors experiments measured here is differences in glial cells?

We are not aware of data that can directly address the ratio of glia to neurons in the dentate gyrus. Our data likely reflect a mixture of neurons (primarily granule cells) and glial cells. To characterize the cellular heterogeneity of our samples, we performed three additional analyses. First, we compared DG expression levels with published data from purified neurons and six distinct glial cell populations from mouse cerebral cortex⁵. Spearman correlation of the expression levels in DG with the purified cell types shows DG is most similar to neurons and least similar to microglia. We added this analysis to Supplementary Figure 2P in the manuscript and include it below.

Supplementary Figure 2P. Correlation of expression levels between the DG tissue and purified brain cell types⁵ reveals expression in DG is most strongly correlated with neurons. *Oligodendrocyte progenitor cells (OPC)*, *newly formed oligodendrocytes (NFO)*, *mature oligodendrocytes (MO)*.

Second, we examined dorsal-ventral expression differences at genes reported by Cembrowski et al. to be strong markers of dorsal or ventral DG granule cells. This analysis showed very strong consistency and is now shown in Supplementar Fig. 2F (and copied below, on page 12 of the response).

Finally, as a third validation, we examined cell type heterogeneity using our methylation data. We correlated gene body mCH, which is strongly associated with gene expression, to previously published neuronal (NeuN+) and non-neuronal (NeuN-) whole-genome bisulfite sequencing samples from a 7-week-old male mouse and 12-month-old female mouse frontal cortex⁶ (Figure 4). The results again showed that DG samples are more strongly correlated with the neuron samples than with the non-neuronal samples. Together, these results suggest that the dominant signal in our data is from neurons and that the effects we report in our study are largely driven by neuronal populations.

Figure R1. Correlation of gene body mCH between the DG and purified NeuN+ and NeuN- samples⁶ shows DG is more similar to neurons than the glia.

In there anything known on different cellular composition in the dorsal and ventral poles that may explain why a whopping 50% of all genes are differentially expressed? Is a 20% DE cutoff with 0.05 FDR a useful cutoff in light of potential heterogeneity issues?

In our original submission we did not impose any fold-change cutoff for the genes we reported as differentially expressed between dorsal and ventral DG, which led to ~50% of genes being detected as DE with statistical significance. We agree that it is potentially misleading to report genes with a small effect size and low expression level, and so we have revised the manuscript to report a more limited set of genes with a moderate or strong expression difference ($\geq 20\%$ or $\geq 100\%$ fold change, respectively). We have also included a cutoff on the absolute expression level (TPM >1 in at least 3 samples). The new text is:

Over 28% of genes expressed in the DG were differentially expressed by region (3,497 out of 12,247 genes; false discovery rate (FDR) < 0.05 , TPM >1 , fold-change $> 20\%$, **Fig. 1C**; Table S2), including 244 genes (2%) with >2 -fold difference in expression.

We also updated Figure 1C (shown below) to show the number of regionally differentially expressed genes as a function of the threshold for mRNA expression difference (fold-change).

In contrast with these results for the difference between dorsal and ventral DG, analysis of differential expression from a previous study of cortical cell types (Excitatory vs. PV- or VIP-positive interneurons) showed 4.7-fold more strongly DE genes (using a cutoff >2 -fold differential expression)⁷; this is shown in S2H and mentioned in the Discussion.

The dorsal and ventral poles of the DG are distinct structures in terms of function and connectivity⁸. Although the cell types in the dorsal and ventral DG are not considered distinct, previous work demonstrated substantial differences in gene expression even within a uniform cell type (DG granule cells)⁹; such differences are also observed for other hippocampal excitatory neuron cell types¹⁰. Our findings are consistent with these results (more on this below), but they go beyond earlier studies in providing a comprehensive, quantitative assessment of the full extent of differential expression. As described above, this is made possible by the large sample size and high technical quality of our data.

Figure 1C. The cumulative number of genes differentially expressed in dorsal vs. ventral DG (FDR < 0.05) as a function of the minimum expression difference cutoff. Here we

consider all genes with >10 RNA-Seq read counts in ≥2 samples (solid lines), or with TPM > 1 in ≥3 samples (dashed).

The authors state that differences are consistent with increased neurogenesis in the dorsal pole. Is this increased neurogenesis expected to also increase differences in heterogeneity?

We agree with the reviewer that multiple factors may contribute to the observed dorsal-ventral expression differences, including differences in expression within mature DG granule cells and differences in the cell type composition, e.g. the proportion of immature neurons. There may also be a contribution from other cell types, such as mossy cells or glia, although we believe that our gene expression data mainly reflects dentate granule neurons (see above, pages 6-7). As the reviewer notes, increased neurogenesis in the dorsal DG could lead to a different proportion of immature neurons in the population. However, the steady state proportion also depends on the rate of neuron death and other developmental factors. With the current data, we cannot distinguish cell type heterogeneity-related expression differences from changes in expression within a cell type. We paste below text from the Limitations section of the manuscript that discussed this limitation:

Although our data are unprecedented in resolution and sample size, there are still some challenges to identifying the source of transcriptional and methylation changes in tissue from a heterogeneous and dynamic cell population like the DG. For example, we cannot distinguish between changes in DNA methylation occurring in a stable population of mature neurons, and changes to the proportion of immature and newborn neurons due to increased neurogenesis. Neurons in all stages of the maturation process coexist within the adult DG, and our data represent a mixture of signals from stem cells and immature and mature neurons. Similarly, the dorsal-ventral differences in DNA methylation could be driven by differences in cell type composition between the two regions, or a discrete or graded difference between the DG neurons in the dorsal and ventral poles. Future studies, including single cell assays, could address these limitations and better characterize methylation in maturing newborn neurons and adult DG neurons¹¹⁻¹³.

This is also relevant for the findings described in the DNA methylation analysis. For instance the authors describe a huge skewing of hypomethylation in DMRs in dorsal versus ventral (23000 regions vs 500). They also find less non cg methylation in dorsal vs ventral. Their explanation of an asymmetric role for DNA methylation in region specific gene regulation is a typical example of the obscurism seen throughout the manuscript which cannot be more vague. Is it not more likely that differences in heterogeneity explain why in one region most of the methylation signal is higher? The authors actually describe this effect at the top of page 9 (lower overall methylation dorsally). If a region becomes more heterogeneous (for instance due to neurogenesis) all signals from specific cell types become diluted. On that note and similar to the RNA analysis, do the authors believe that the thresholds used in their analysis are workable in light of differences in heterogeneity (15% > difference)?

In the discussion after an elaborate speculation on the molecular differences between dorsal and ventral differences and the impact of EE on epigenetic signatures the authors seem to finally acknowledge that heterogeneity may be a limiting factor in their work but then revert back to speculating and continue to claim that they assess the environmental effects on DG neurons.

As we write in the Discussion, cell type heterogeneity may be affected by environmental enrichment and this could be one possible explanation for some of the transcriptional and epigenetic differences we observe. This concept is illustrated and discussed in Fig. 5 of the paper. There is likely a complex network of relationships among (1) epigenetic and transcriptional impacts of EE on existing cells (both neurons and radial glia like progenitors); (2) the secondary effect of these changes on the regulation of neurogenesis, and thereby on the heterogeneity (or, more precisely, on the mixture of cell types); and finally, (3) the impact of environmental enrichment on the epigenome and transcriptome of newborn neurons themselves. Unfortunately, dissecting the contribution of cellular heterogeneity would require single cell sequencing approaches^{14–16}, which are still very in development (including by some authors on this paper¹⁶), but which have not yet been used to examine experience-dependent transcriptional and epigenetic programs. Although we cannot necessarily distinguish between changes to existing cells vs. changes to the relative population of different cell types, we would argue that in either case the changes we observe represent environmental impacts on DG cells.

Regarding the threshold for differential methylation, there is no universally agreed upon standard in the field. We used a published DMR detection software package, DSS¹⁷, to call DMRs, and we set the parameter “delta” to 15% to require that each individual CG site contributing to a DMR has $\geq 15\%$ difference in methylation. Many DMRs actually have methylation differences substantially greater than this minimum; to show this we have added a plot showing the cumulative number of DMRs as a function of threshold (Supp. Fig. 3H, shown below). Since the threshold of 15% is applied to all the sites within a DMR, the average difference in mCG is actually $>20\%$ for almost all of the DMRs in our list. The mean methylation difference across all DMRs is $26.2\% \pm 4.5\%$ s.d. There are thousands of DMRs with $>30\%$ methylation difference, and hundreds with $>40\%$ difference. Notably, no matter what threshold we choose, we find that the number of DMRs hypo-methylated in dorsal DG (dDG<vDG) is always ~ 10 -fold larger than the number hypo-methylated in ventral DG (vDG<dDG). Hence the regional-specificity we report is apparent regardless of DMR threshold. This is an important point and we appreciate the challenge from the reviewer on this issue.

To clarify these points, we have added Supplementary Fig. 3H and included the mean methylation difference in the text.

Supplementary Figure 3H: Number of DMRs as a function of minimum methylation difference.

With the low thresholds the authors use for looking at differences in methylation I'm honestly not sure what they measure in their analysis. The authors could focus on more stringent changes (new dorsal methylation regions that could say something about the new cells generated dorsally). The authors also do not contrast their data to the existing datasets other than comparing the correlation scores of replicates to argue their data is better.

We directly compare our data with a published RNA-Seq data set (Hippo-Seq) for dorsal and ventral DG granule cells in Fig. S2E,F⁹. We have now extended this comparison by adding a plot (Fig. S2F, below) that compares the fold-change in expression between dorsal and ventral DG in our data set vs. Hippo-Seq. This plot highlights all of the genes identified in Hippo-Seq data as dorsal-enriched (e.g. *Lct*) or ventral enriched (e.g. *Nr2f2*). Every one of these genes shows a consistent direction of differential expression in our data set. As expected, due to the fact that our data come from mixed tissue, and potentially also because of the lower noise level of our expression estimates, the absolute magnitude of the fold-change is lower in our data set compared with the microdissected DG granule cells.

Supplementary Fig. 2F: Comparison of regional differences in expression between the present data set (DG tissue) and purified granule cells from Hippo-Seq⁹ shows highly consistent differential expression for markers of dorsal and ventral DG granule cells.

Is there anything to the common changes in dorsal and ventral between SH and EE and the data previously published?

To our knowledge, our study is the first to directly compare EE vs. SH in tissue isolated from the dorsal and ventral DG. Consequently, we cannot make a direct comparison with previous studies. We do, however, compare with three previous studies that examined EE differences in the hippocampus (see text below)¹⁸⁻²⁰.

Brain-derived neurotrophic factor (BDNF) is upregulated at the mRNA level in mouse hippocampus following 3-4 weeks of exposure to EE¹⁹, while EE-induced adult neurogenesis was blocked in a heterozygous knockout (Bdnf^{f/-})¹⁸. Similarly, mRNA for vascular endothelial growth factor (VEGF) is upregulated in hippocampus upon exposure to EE, and manipulations that increase or decrease VEGF levels cause corresponding increases and decreases in neurogenesis²⁰. We did not detect differential expression of Bdnf or Vegf in the dorsal or ventral DG, suggesting these factors may be upregulated in other hippocampal regions.

Do DMRs explain differences observed in specific cell types in the previous studies? Are there key known factors to look at? Why is this not extensively cross compared?

Other than Hippo-Seq, there are no previous studies that, to our knowledge, have examined differences in DG gene expression between the dorsal and ventral poles. The signal in the Hippo-Seq data isn't of sufficient quality for us to make a direct comparison. We believe making comparisons within our own samples is the cleanest and most controlled approach we can take. Therefore, we extensively compared our DMRs with gene expression differences in our own samples (Fig. 2D-G, Supplementary Fig. 3B-D,F-G). Although methylation is generally negatively correlated with expression in the brain⁶, our results illustrate that this relationship at the gene level is very complex. Consequently, we would not expect a direct match between genes that show DMRs and genes that are differentially expressed.

All in all I find the data extensive and valuable as a resource albeit for a specialized audience. However I believe the analysis is trying to obscure that nothing is really yet found. I believe a more honest assessment of the data would enable the community to use the resource for what it is instead of what the authors are trying to make it and perhaps a different focus to the analysis may very well uncover a story that is fit for publication in a journal with a wide readership.

We thank the reviewer for their comments. We agree that our dataset will be a great resource to others in the field, and we hope that our comments above have helped address their concerns.

References

1. Artola, A. *et al.* Long-lasting modulation of the induction of LTD and LTP in rat hippocampal CA1 by behavioural stress and environmental enrichment. *Eur. J. Neurosci.* **23**, 261–272 (2006).
2. Hodes, G. E., Walker, D. M., Labonté, B., Nestler, E. J. & Russo, S. J. Understanding the epigenetic basis of sex differences in depression. *J. Neurosci. Res.* **95**, 692–702 (2017).
3. Hodes, G. E. *et al.* Sex Differences in Nucleus Accumbens Transcriptome Profiles Associated with Susceptibility versus Resilience to Subchronic Variable Stress. *J. Neurosci.* **35**, 16362–16376 (2015).
4. Kempermann, G., Kuhn, H. G. & Gage, F. H. More hippocampal neurons in adult mice living in an enriched environment. *Nature* **386**, 493–495 (1997).
5. Zhang, Y. *et al.* An RNA-sequencing transcriptome and splicing database of glia, neurons, and vascular cells of the cerebral cortex. *J. Neurosci.* **34**, 11929–11947 (2014).
6. Lister, R. *et al.* Global epigenomic reconfiguration during mammalian brain development. *Science* **341**, 1237905 (2013).
7. Mo, A. *et al.* Epigenomic Signatures of Neuronal Diversity in the Mammalian Brain. *Neuron* **86**, 1369–1384 (2015).
8. Fanselow, M. S. & Dong, H.-W. Are the dorsal and ventral hippocampus functionally distinct structures? *Neuron* **65**, 7–19 (2010).
9. Cembrowski, M. S., Wang, L., Sugino, K., Shields, B. C. & Spruston, N. Hipposeq: a comprehensive RNA-seq database of gene expression in hippocampal principal neurons. *Elife* **5**, e14997 (2016).
10. Cembrowski, M. S. *et al.* Spatial Gene-Expression Gradients Underlie Prominent Heterogeneity of CA1 Pyramidal Neurons. *Neuron* **89**, 351–368 (2016).
11. Lacar, B. *et al.* Nuclear RNA-seq of single neurons reveals molecular signatures of activation. *Nat. Commun.* **7**, 11022 (2016).

12. Angermueller, C. *et al.* Parallel single-cell sequencing links transcriptional and epigenetic heterogeneity. *Nat. Methods* **13**, 229–232 (2016).
13. Habib, N. *et al.* Div-Seq: Single-nucleus RNA-Seq reveals dynamics of rare adult newborn neurons. *Science* **353**, 925–928 (2016).
14. Habib, N. *et al.* Div-Seq: Single-nucleus RNA-Seq reveals dynamics of rare adult newborn neurons. *Science* **353**, 925–928 (2016).
15. Lacar, B. *et al.* Nuclear RNA-seq of single neurons reveals molecular signatures of activation. *Nat. Commun.* **7**, 11022 (2016).
16. Luo, C. *et al.* Single-cell methylomes identify neuronal subtypes and regulatory elements in mammalian cortex. *Science* **357**, 600–604 (2017).
17. Feng, H., Conneely, K. N. & Wu, H. A Bayesian hierarchical model to detect differentially methylated loci from single nucleotide resolution sequencing data. *Nucleic Acids Res.* (2014).
18. Rossi, C. *et al.* Brain-derived neurotrophic factor (BDNF) is required for the enhancement of hippocampal neurogenesis following environmental enrichment. *Eur. J. Neurosci.* **24**, 1850–1856 (2006).
19. Kuzumaki, N. *et al.* Hippocampal epigenetic modification at the brain-derived neurotrophic factor gene induced by an enriched environment. *Hippocampus* **21**, 127–132 (2011).
20. Cao, L. *et al.* VEGF links hippocampal activity with neurogenesis, learning and memory. *Nat. Genet.* **36**, 827–835 (2004).

REVIEWERS' COMMENTS:

Reviewer #1 (Remarks to the Author):

The manuscript has been revised to fully address my comments. I have no additional concerns.

Reviewer #3 (Remarks to the Author):

In their revisions the authors have addressed most of my concerns considering data representation save a few annoyances. While the basis for the differences seen in their samples remains unclear and clear biological insight is not yet found the data does represent a valuable and resource.

Nevertheless in figure 2P the authors try to make the argument that neurons are closer to DG than glial cells. Why do they try this. Look at the scale bar, (runs between 0.8 and 1.0). The statement that DG is least similar to microglia is again trying to play hide the ball with the data. Microglia are immune cells and represent around 1% of glial cells and most glia (70%) are in fact oligodendrocytes. The difference between oligodendrocytes and neurons compared to DG is marginal (guesstimating from the figure .87 to .89?). Again this is not a major deal as the authors now acknowledge heterogeneity as an issue in the discussion now but it's simply annoying. Where do the environmental enrichment samples fall in this correlation analysis, do they shift towards one end?, this should be shown? The effect is stronger in figure R1 which could also come from differences in methylation levels between neurons and glial cells. Again where are the EE DG samples in this analysis? My point here is; the notion that heterogeneity could play a role here does not diminish the value of the data as a resource but should be fairly and clearly acknowledged.

Reviewer 3

In their revisions the authors have addressed most of my concerns considering data representation save a few annoyances. While the basis for the differences seen in their samples remains unclear and clear biological insight is not yet found the data does represent a valuable resource.

We thank the reviewer for evaluating and accepting our revisions. We address the remaining concerns below.

Nevertheless in figure 2P the authors try to make the argument that neurons are closer to DG than glial cells. Why do they try this. Look at the scale bar, (runs between 0.8 and 1.0). The statement that DG is least similar to microglia is again trying to play hide the ball with the data. Microglia are immune cells and represent around 1% of glial cells and most glia (70%) are in fact oligodendrocytes. The difference between oligodendrocytes and neurons compared to DG is marginal (guesstimating from the figure .87 to .89?). Again this is not a major deal as the authors now acknowledge heterogeneity as an issue in the discussion now but it's simply annoying.

See final response below.

Where do the environmental enrichment samples fall in this correlation analysis, do they shift towards one end?, this should be shown? The effect is stronger in figure R1 which could also come from differences in methylation levels between neurons and glial cells. Again where are the EE DG samples in this analysis?

To examine whether enriched environment causes a shift in cellular heterogeneity, we correlated our RNA-seq and DNA methylation data from the DG separately for SH and EE animals with the expression and methylation profiles known cell type populations (Fig. R1). Results do not show any substantive differences between SH and EE animals. However, this is a very coarse analysis that likely doesn't have sufficient sensitivity to detect such smaller effects. Future work that can control for cellular heterogeneity is required to best elucidate the effects of EE.

Figure R1. (A) Correlation of expression levels between the DG tissue for SH and EE samples with purified brain cell types⁵ reveals expression in DG is most strongly correlated with neurons. *Oligodendrocyte progenitor cells (OPC)*, *newly formed oligodendrocytes (NFO)*, *mature oligodendrocytes (MO)*. (B) Correlation of gene body mCH between the DG for SH and EE samples with purified NeuN+ and NeuN- samples shows DG is more similar to neurons than the glia.

My point here is; the notion that heterogeneity could play a role here does not diminish the value of the data as a resource but should be fairly and clearly acknowledged.

We agree with the review that the cellular heterogeneity should be fairly and clearly acknowledged in our study. Therefore, we have further qualified the discussion of our results to include the following text:

Here we attempted to better understand the heterogeneity of our tissue by correlating our RNA-seq data with known neuron type transcriptional profiles. Although the strongest correlation between our dDG and vDG bulk tissues was with neurons ($r=.89$), we also found substantial correlations with gene expression patterns in other cell types. Thus, it remains difficult to determine to what extent regional differences and EE-induced changes in cellular heterogeneity may account for our results.